# An Attention-based Approach for Bayesian Optimization with Dependencies

## Abstract

Bayesian Optimization (BO) is a sample-efficient method for optimizing black-box problems that are expensive to evaluate. The canonical BO is conducted in search spaces where hyperparameters are independent and the dimension of configurations remains fixed. However, different algorithms typically require their own distinct hyperparameters in practice, thereby yielding a hierarchical search space structure. Such a nested configuration challenges the direct application of Bayesian optimization, as it obscures the independence assumptions made in the standard Bayesian optimization formulation. In this paper, we propose a structure-aware embedding and an attention-based Deep Kernel Gaussian Process to capture the response surface in such conditional search spaces. By endowing the surrogate model with context on the conditional structure, our approach facilitates Bayesian optimization in navigating nested hyperparameter configurations. Empirical results on both a tree-structured simulation benchmark and several real-world benchmarks demonstrate that our proposed approach improves the efficacy and efficiency of BO in conditional search spaces.

## 1 Introduction

Bayesian Optimization (BO) (Mockus et al., 1978; Balandat et al., 2020) is a powerful and efficient global optimizer for expensive black-box functions, which has gained increasing attention in AutoML systems and achieved great success in a number of practical application fields in recent years (Calandra et al., 2016; González et al., 2015; Lizotte et al., 2007; Martinez-Cantin et al., 2007). Considering a black-box function $f\colon \chi \to \mathbb{R}$ defined on a search space $\chi$, BO aims to find the global optimal configuration

$$x^* = \arg\min_{x \in \chi} f(x). \tag{1}$$

The sequential Bayesian optimization procedure contains two key steps: (1) BO seeks a probabilistic surrogate model to capture the distribution of the black-box $f$ given n noisy observations $y_i = f(x_i) + \epsilon, i \subset 1, ..., n, \epsilon \sim \mathcal{N}(0, \sigma)$. (2) Suggest the next query $x_{n+1}$ by maximizing an exploit-explore trade-off acquisition function $\alpha(x)$. The most common choice of the surrogate models is Gaussian Process (GP) (Snoek et al., 2012; Seeger, 2004) due to its generality and good uncertainty estimation. As to acquisition functions, the common choice for GP-based BO is the Expected Improvement (EI) (Mockus, 1994), which balances the exploration and exploitation and provides a theoretical regret bound.

In the traditional BO setting, the search space $\chi$ is flat where the configuration has the same dimensions and similar structure [1]: $x \in \chi \subset \mathbb{R}^d$, where $d$ is the dimension. However, in many practical machine learning scenarios, such as Combined Algorithm Selection and Hyperparameter optimization (CASH) problem (Thornton et al., 2013; Levesque et al., 2017), the search space $\chi$ is hierarchical and consists of multiple subspaces with different structure of configurations and even different dimensions. In such a setting, the hierarchical space can be decomposed into a series of flat subspaces: $\chi = \chi^1 \cup \chi^2 \cup ... \cup \chi^n$ and configurations in the same subspace has the same structure: $x^i \in \chi^i \subset \mathbb{R}^{d^i}$.

---

[1] In this paper, we define the structure of a configuration, which contains two aspects: 1. dependencies between every pair of hyperparameters; 2. semantic information of each hyperparameter, e.g. the hyperparameter "learning rate" has similar semantic information in XGBoost and DNN models.

A straightforward strategy is to build separate surrogate models for each subspace independently. However, this approach requires considerable search cost as it treats each subspace as isolated. Additionally, it fails to leverage potential relationships between subspaces, which could provide useful information to guide the optimization. Nguyen et al. (2020) utilize Thompson Sampling (Thompson, 1933; Snoek et al., 2014), a bandits method which has theoretical regret bound, to helps connecting both multi-arm bandit and BO in a unified framework. However, it still suffers from the inefficiency of separate GP models and needs more observations to guarantee performance.

Recent works have proposed using variational autoencoders (VAEs) to transform structured, high-dimensional optimization problems into continuous, low-dimensional spaces that are more amenable to Bayesian optimization techniques (Kusner et al., 2017; Lu et al., 2018; Tripp et al., 2020; Grosnit et al., 2021; Maus et al., 2022). However, these existing methods do not readily extend to search spaces containing both categorical and numerical hyperparameters in a complex, structured relationship. In this paper, we concentrate on building a more powerful and general surrogate model for applying BO in such complex structured but not high-dimensional spaces.

Some recent works (Jenatton et al., 2017; Ma & Blaschko, 2020b) propose to regard the hierarchical spaces as a tree and model the black-box object function through decision trees. Specifically, Jenatton et al. (2017) proposed to separately build a GP model on each flat subspace and introduced a weight vector to integrate the GPs linearly. Another work (Ma & Blaschko, 2020b) assumed that the performance of configurations on different subspaces is independent and proposed an additive covariance function to capture the global response surface of the objective function. However, both works ignore the dependencies of the hyperparameters in configurations and have limitations in practical application due to their linear and additive assumptions.

In this paper, we propose an elegant attention-based BO framework to directly capture the global response surface in the hierarchical search space by a single Deep Kernel Gaussian Process (DKGP) surrogate model. Specifically, we provide a general attention-based encoding method, which can embed the semantics and dependencies information into the configurations sampled from different subspaces. Then we project the configurations from different subspaces into a unified latent space, where the configurations can be comparable and modeled by any standard kernel functions, such as Matérn 5/2 (M52) and squared exponential (SE) kernel function. Our attention-based encoder can deal with variable-length input sequences and capture the global relationships among hyperparameters in a specific configuration. In the acquisition stage, our proposed method can optimize the acquisition function in each flat subspace which could provide batch quires for evaluation.

In conclusion, our contributions can be summarized as follows:

**1) Structure-aware embeddings.** We provide a general encoding method for preserving the semantic and dependency information of each hyperparameter in a configuration, leveraging the prior relationships among hyperparameters during modeling the response surface.

**2) A Unified latent space for configurations with different structures.** With the structure-aware embeddings, we propose a novel attention-based encoder, which is able to capture global relationships among hyperparameters and project the variable-length configurations into a unified latent space.

**3) An Efficient BO framework with deep kernel learning.** Following the idea of deep kernel learning, we utilize our proposed structure-aware embeddings and attention-based encoder to learn a deep kernel for directly capturing the global response surface in the hierarchical space using a single GP. Instead of the state-of-the-art work (Ma & Blaschko, 2020b), our approach relaxes the assumption of the objective function and becomes more general for practical AutoML applications. Moreover, our proposed BO framework can achieve parallel searching and improve efficiency when handling the black-box functions which are very expensive to evaluate.

**4) Strong Performance on multiple benchmarks.** We conduct experiments on a standard tree-structured simulation benchmark, a Neural Architecture Search (NAS) benchmark which is similar to Tan et al. (2019), and several real-world OpenML benchmarks. The experimental results demonstrate the efficiency and efficacy of our proposed approach.

## 2 RELATED WORK

### 2.1 BAYESIAN OPTIMIZATION FOR CONDITIONAL SEARCH SPACE

Sequential Model-based Algorithm Configuration (SMAC) (Hutter et al., 2011) and Tree-structured Parzen Estimator (TPE) (Bergstra et al., 2011) are two early BO methods that can deal with multi-family problems. SMAC utilizes random forest instead of GP as the surrogate model and imputes the inactive dimensions with default values to deal with the conditional space. However, the imputed dimensions would lead to higher-dimension problems and reduce the efficiency during optimization. TPE models two densities to estimate whether the response of a configuration is good or not, which is naturally structure-free and can be directly applied to multi-family problems. However, it ignores the relationship between dimensions and requires more observations to capture the densities effectively.

Compared to SMAC and TPE, GP-based BO gives better uncertainty estimation and shows higher sample efficiency in practical applications. The most straightforward way to leverage GP to solve multi-family problems is building an independent GP in each subspace, however, which totally ignores the meta information that can be shared between different subspaces. Jenatton et al. (2017) proposed a semi-parametric GP method that captures the relationship of GPs via a weight vector. Although the idea of the weight vector can establish a mechanism for sharing information between multiple Gaussian processes, the assumption of linear relationships will limit the effectiveness and generalization of this method. Following this work, Ma & Blaschko (2020b) proposed an Add-Tree covariance function to capture the global response surface of $f$ using a single GP. It gives an additive assumption on the objective function that each vertex of the tree-structure search space is independent, which would be invalid when there are relationships between the hyperparameters in these vertexes. Moreover, the similarity of configurations is only built on their sharing vertexes, which totally ignores the non-shared hyperparameters between configurations during the similarity modeling and could not capture the global meta-features of a configuration.

### 2.2 DEEP KERNEL LEARNING FOR GAUSSIAN PROCESS

The standard approach to fit GPs is to optimize the parameters of the handcrafted kernel function, however, which would lead to sub-optimal performances due to the false assumptions (Cowen-Rivers et al., 2022). The idea of deep kernel learning (Wilson et al., 2016) is to learn the kernel function using a neural network $\phi$ to transform the configuration $x$ to a latent representation that serves as the input of the kernel, which facilitates learning the kernel in a suitable space. Specifically, the kernel function is shown as:

$$k_{deep}(x, x^{'}|\theta, \omega) = k(\phi(x, \omega), \phi(x^{'}, \omega)|\theta), \tag{2}$$

where $\omega$ represents the weights of the deep neural network $\phi$ and $\theta$ represents the parameters of the handcrafted kernel function, e.g., Matérn 5/2 function. All these parameters can be jointly estimated by maximizing the marginal likelihood (Wistuba & Grabocka, 2021).

### 2.3 ATTENTION

In the field of Natural Language Processing (NLP), the transformer model (Vaswani et al., 2017) is a pioneering work, which uses the attention module to model the global relationship between different words in a sequence, such as a sentence and paragraphs. In many later practices and papers, the effectiveness of the attention module was verified and applied in many fields (Lin et al., 2022). For an input sequence of $N$ words, the $d_k$-dimensional embeddings of words plus the corresponding positional embeddings are fed into a stacked attention module. In the attention mechanism, the packed matrix representation of the query $Q \in \mathbb{R}^{N \times d_k}$, the key $K \in \mathbb{R}^{N \times d_k}$ and the value $V \in \mathbb{R}^{N \times d_k}$ are fused through $\text{Attention}(\mathbf{Q}, \mathbf{K}, \mathbf{V}) = \text{softmax}\left(\frac{\mathbf{Q}\mathbf{K}^{\top}}{\sqrt{D_k}}\right)\mathbf{V} = \mathbf{AV}$. The attention matrix $A$ contains the similarity between each pair of words, which makes the output feature of a word a fusion of the feature of each word in the whole sequence. In this paper, this mechanism is employed to model the relationship among hyperparameters in a tree-structured search space, where the sampled configuration can be viewed as a sequence of hyperparameters.

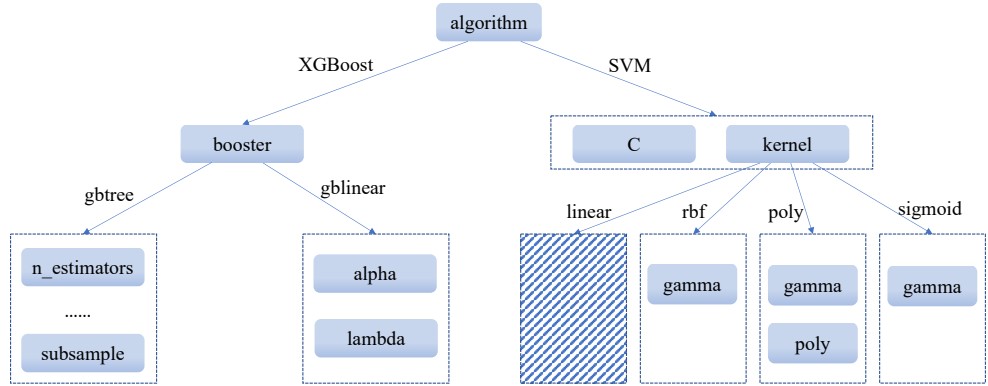

Figure 1: An example of the tree-structured search space for the tabular classification tasks, which contains two popular algorithms and their distinct hyperparameters. When the kernel is linear for the SVM model, the shaded box indicates that there are no hyperparameters in this case.

## 3 CONDITIONAL SEARCH SPACE MODELING

In black-box optimization applications, possibly, there is a conditional structure in the search space. Given a task with the corresponding datasets, we need to choose the best algorithm for it from multiple model options. However, not only do different algorithms require different hyperparameters, but also different microstructures have different value ranges for the same hyperparameter, resulting in dependencies among hyperparameters. For example, given a classification task on a tabular dataset, XGBoost (Chen & Guestrin, 2016) and SVM (Cervantes et al., 2020) can be candidates with different hyperparameters, such as booster type in XGBoost and kernel in SVM (see Fig. 1). Also, the various boosters require various hyperparameters such as the number of estimators in the former but not in the latter. When searching the neural networks in CNN, the ranges of the appropriate number of channels always alter across different layers.

Here, we give a loose assumption that such a conditional structure is a tree structure or a combination of several tree structures. To avoid many repetitions and to simplify the notation, in this paper, we use a tree structure referring to multiple trees, since embedding methods and our method can also handle this case. Inspired by Ma & Blaschko (2020b), we define the search space $\chi$ as a tree structure $\mathcal{T} = (P, E)$, where one node $p \in P$ refers to one hyperparameter with associated range, type and value (if sampled), and $e \in E$ refers to the dependency relationship between a node $p$ and the father node $p \uparrow \in P$. To unify the notation, for those root nodes without a father, we assign a virtual father vertex with a fixed embedding. Here, the ancestor nodes represent categorical variables, whereas leaf nodes that have no children, can be of various types including *integers*, *floats*, and *categories*. From the data structure view of this tree structure, since each node has only one parent node or not, we can serialize the tree by storing nodes and their corresponding father nodes, as shown in Section 4.1. The tree structure search space $\chi$ can be extended as a set of subspaces $\{\chi^i\}$ without dependency, where each subspace is a path from the root to a leaf node. After sampling from the search space $\chi$, we group the configurations $\mathbf{X}$ by each subspace $\chi^i$ as $\mathbf{X}^i = \{x^i_j\}$, where $i = 1, 2, ..., n$ and $j = 1, 2, ..., N^i$, $N^i$ represents the number of points belongs to the subspace $\chi^i$. A configuration $x^i_j$ is a set of hyperparameters $\left\{p_k^{x^i_j}\right\}$ with specific values, where $k = 1, 2, ..., d^i$, $d^i$ means the dimension of the subspace $\chi^i$.

## 4 DEEP KERNEL LEARNING WITH ATTENTION

GP serves as a good surrogate model due to its good uncertainty estimation and sample efficiency, however, to the best of our knowledge, there is no suitable hand-crafted kernel function capable of modeling the similarity between these configurations from different subspaces which have different dimensions and dependencies. In this paper, we seek to build a global GP in a hierarchical conditional search space to achieve higher efficiency and efficacy. In order to achieve our purpose, we need to project these configurations $x^i, i = 1, 2, ..., n$, into a unified latent space $\mathcal{Z} \subset \mathbb{R}^d$, enabling the

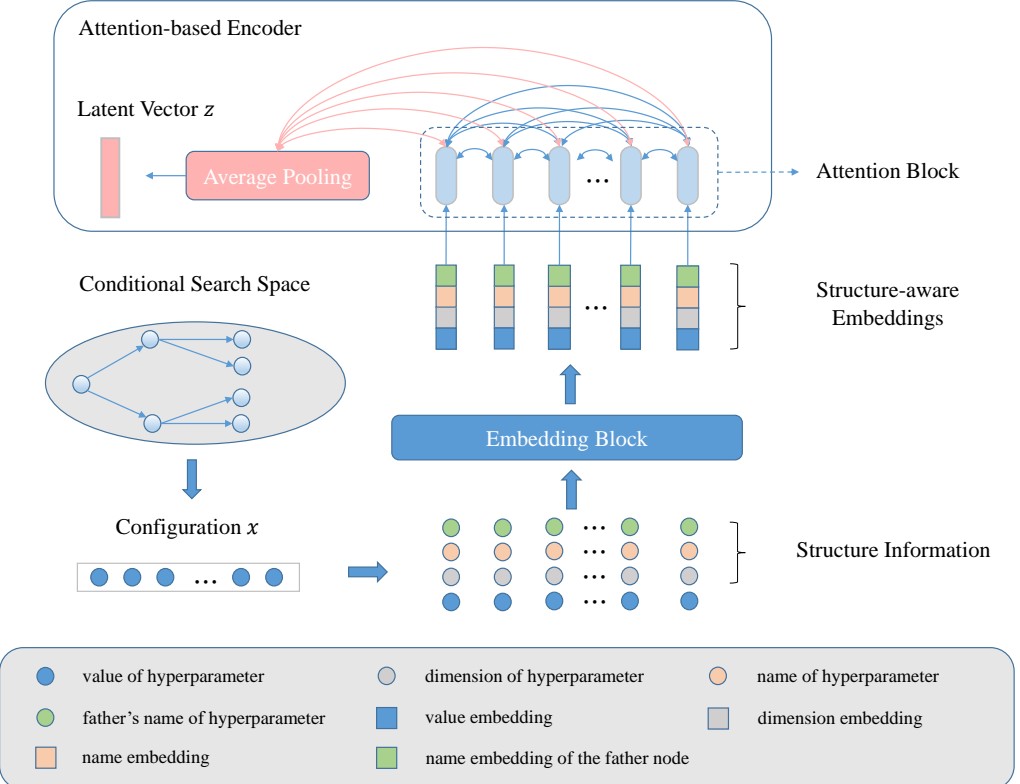

Figure 2: The framework of our proposed AttnBO.

configurations in different spaces to be comparable and modeled directly by a single kernel. Although the idea of deep kernel learning provides a way to learn suitable kernel functions in different situations, finding a deep neural network that can deal with variable-length configurations and capture the global relationships between hyperparameters in these configurations is unprecedented.

Recently attention-based models, which have the ability to capture the global relationship between words in variable-length sequences, have achieved great success in Natural Language Processing (NLP). In the context of NLP, the words need to be embedded into vectors and then fed to the attention-based models with their positional encoding, which contains the positional information in sequences, to introduce the ordering relationships between words. In the setting of our problem, the configurations from different subspaces can be viewed as variable-length sequences of hyperparameters, with each hyperparameter representing a token. However, unlike in NLP, the position of hyperparameters in a configuration need not be taken into account, and the dependencies between them require consideration. Thus, we propose a dependencies-aware embedding method to introduce the dependencies between hyperparameters during modeling. And then, with these dependencies-aware embeddings, we utilize an attention-based encoder to capture the meta feature of the configurations which consider the dependencies and global relationships between their hyperparameters and project these configurations into a unified latent space $\mathcal{Z} \subset \mathbb{R}^d$.

Based on the dependencies-aware embeddings and attention-based encoder, we propose an efficient BO framework, named AttnBO, which can build a global Deep Kernel GP in a complex conditional search space. The outlook of the proposed framework is shown in Fig. 2.

## 4.1 STRUCTURE-AWARE EMBEDDINGS

The hyperparameters in configurations sampled from different subspaces have different semantics and relationships, which the surrogate model should be aware of for capturing the hierarchical response surface. Suppose we have a tree-structured search space that is composed of two subspaces, the hyperparameters of configurations from the two subspaces, which are $(p_1, p_2, p_3, p_4)$ and $(p_1, p_5, p_6)$ respectively, obviously have different semantics and relationships, which makes the vectors containing only hyperparameter values cannot be used directly to fit the surrogate model. Previous BO works

aligned the configurations in all subspaces by imputing inactive hyperparameters with some default values, however, which would lose the space-specific dependencies and lead to a higher dimension.

Inspired by this point, we assign each hyperparameter an embedding that contains semantic and dependency information, instead of only considering the value of the hyperparameter as in traditional BO methods. In a tree-structured conditional search space, restoring the structure of the tree only requires identifiers for each node and its father. Thus, as Fig. 2 shows, we can encode a hyperparameter $p_k^{x_j^i}$ to a structure-aware embedding with four elements: **1) the name embedding** $name\_emb(p_k^{x_j^i})$**, 2) the name embedding of its father** $name\_emb(p_k^{x_j^i} \uparrow)$**, 3) the dimension embedding** $dim\_emb(p_k^{x_j^i})$ **and 4) the value embedding** $value\_emb(p_k^{x_j^i})$. We take into account the dimension information because some hyperparameters, such as the number of hidden units in a multi-layer deep neural network, may be represented as lists. In this setting, configurations $x_j^i, j = 1...N^i$ in the same space $\chi^i$ have the same name embeddings, father name embeddings and dimension embeddings, which will be different in different space. We concatenate these four embeddings as the representation of a hyperparameter $p_k^{x_j^i}$:

$$emb(p_k^{x_j^i}) = concat(name\_emb(p_k^{x_j^i}), name\_emb(p_k^{x_j^i} \uparrow),$$
$$dim\_emb(p_k^{x_j^i}), value\_emb(p_k^{x_j^i})), k = 1, 2, ..., d^i, \tag{3}$$

where $d^i$ represents the dimension of the flat subspace $\chi^i$, With such embeddings, we transform each configuration into a vectorial representation that contains space-specific semantic information and dependencies, which can be used for training a structure-aware surrogate model to capture the hierarchical response surface. The full embedding of a configuration is represented as:

$$emb(x_j^i) = \left\{ emb(p_1^{x_j^i}), emb(p_2^{x_j^i}), ..., emb(p_{d^i}^{x_j^i}) \right\}. \tag{4}$$

To demonstrate the effectiveness of the structure-aware embedding, we conducted an ablation study on these embeddings, and the experimental results can be found in Appendix E.

## 4.2 Attention-based Encoder for Deep Kernel Gaussian Process

Although we have embedded semantic information into the configuration, the configurations $\mathbf{X} = \left\{ x_j^i | i = 1, 2, ..., n, j = 1, 2, ..., N^i \right\}$ in different subspaces, where $N^i$ represents the numbers of observations in search space $\chi^i$, are of varying lengths and have different hyperparameters, which is still a challenge to project them into a unified latent space $\mathcal{Z} \subset \mathbb{R}^d$. Therefore, we introduce an attention model that can handle variable-length sequences and capture global relationships into our framework to solve this problem. With an attention-based encoder $\phi : \chi \to \mathcal{Z}$, we can exploit the meta feature of the configurations which consider the dependencies and relationships between their hyperparameters. And then a GP can be built on this latent space $\mathcal{Z}$ with a standard kernel function, e.g., Matérn 5/2 kernel. To demonstrate this ability of our method, we give the visualization of the attention map among the hyperparameters in Appendix D.

We adopt the deep kernel learning framework to learn the weights of the embeddings, attention-based encoder, and the parameters of the kernel function jointly by maximizing the log marginal likelihood:

$$\log p(\mathbf{y}|\mathbf{X}, \theta, \omega) \propto -(\mathbf{y}^T \mathbf{K}_{deep}^{-1} \mathbf{y} + \log(|\mathbf{K}_{deep}|)), \tag{5}$$

where $\mathbf{y} = \left\{ y_j^i | i = 1, 2, ..., n, j = 1, 2, ..., N^i \right\}$ represents the noisy response of all configurations and $\omega_1, \omega_2$ are two subsets of $\omega$ which represent the weights of the embeddings and the attention-based encoder respectively. According to eq. 2, the deep kernel matrix is as follow:

$$\mathbf{K}_{deep} = k_{deep}(\mathbf{X}, \mathbf{X}|\theta, \omega) + \sigma^2 \mathbf{I} \tag{6}$$
$$= k(\phi(emb(\mathbf{X}, \omega_1), \omega_2), \phi(emb(\mathbf{X}, \omega_1), \omega_2)|\theta) + \sigma^2 \mathbf{I} \tag{7}$$

.

### 4.3 BAYESIAN OPTIMIZATION WITH THE ATTENTION-BASED DKGP

Consider a black-box function with nosiy observations $y_i = f(x_i) + \epsilon, i \subset 1, ..., n, \epsilon \sim \mathcal{N}(0, \sigma)$, we have a dataset $D$ of $N$ noisy observations in a conditional space $\chi$ that has $n$ flat subspaces $\left\{ \chi^1 \cup \chi^2 \cup ... \cup \chi^n \right\}$, $N = \sum_{i=1}^{n} N^i$, $D = \left\{ D^1, D^2, ..., D^n \right\}$, where $D^i$ means all observations $\left\{ (x_j^i, y_j^i) | j = 1, 2, ..., N^i \right\}$ in subspace $\chi^i$. The predictive posterior distribution of the objective function $f$ at $x^*$ is as follow:

$$f_* | \mathbf{X}, \mathbf{y}, x_* \sim \mathcal{N} \left( \overline{f}_*, var(f_*) \right) \quad (8)$$

where

$$\overline{f}_* = k_{deep}(x_*, \mathbf{X}) \mathbf{K}_{deep}^{-1} \mathbf{y}, \quad (9)$$

$$var(f_*) = k_{deep}(x_*, x_*) - k_{deep}(x_*, \mathbf{X}) \mathbf{K}_{deep}^{-1} k_{deep}(\mathbf{X}, x_*), \quad (10)$$

and the deep kernel matrix $\mathbf{K}_{deep}$ can be founded in eq. 2. In this paper, we use the Matérn 5/2 kernel function to accommodate the DKGP model and adopt EI acquisition function to choose the next query. During the acquisition stage, we optimize EI on each subspace and find the most valuable configurations to query in each subspace, enabling parallel Bayesian optimization on the objective functions. Under the sequential BO setting, we choose the configuration that has the highest value of the acquisition function among all subspaces. The detailed procedure of the algorithm can be found in Algorithm 1.

## 5 EXPERIMENTS

To demonstrate the efficiency and efficacy of AttnBO, we conduct experiments on multiple benchmarks, including a simulation benchmark used in Jenatton et al. (2017); Ma & Blaschko (2020b), a NAS benchmark whose search space is similar to Tan et al. (2019) evaluated on cifer-10 dataset, and several real-world benchmarks on OpenML. For the simulation benchmark, we follow the setting of Ma & Blaschko (2020b), which has three binary decision variables $x_1, x_2, x_3$, two shared variables $r_8, r_9$ bound in [0, 1], and four non-shared numerical variables $x_4, x_5, x_6, x_7$ bounded in [-1, 1].

Following the solid work Tan et al. (2019) in the NAS field, we set an optimization problem in a complex search space which includes a minimum of 29 and a maximum of 47 hyperparameters depending on different conditions —- the number of the blocks ranging from 4 to 7. There are both categorical and continuous hyperparameters in this NAS space, and the candidate will be evaluated on CIFAR-10 dataset after 100 training epochs. The details of the settings of this search space can be found in Appendix C.3.

For the OpenML benchmarks, we design two hierarchical search spaces for SVM and XGBoost respectively, which are two popular machine-learning models for tabular data. The SVM search space has four hyperparameters: 1) C {type: float, range: [0.001, 1000]}, 2) kernel {type: choice, range: {linear, poly, rbf, sigmoid}}, 3) degree {type: int, range: [2, 5]}, 4) gamma {type: float, range: [0.001, 1000]}. Different kernels need kernel-specific hyperparameters, which leads to a hierarchical space. The hyperparameter gamma is valid except when the kernel is set to linear. And the hyperparameter degree is only valid when the kernel is set to poly. As to XGBoost, we set a categorical hyperparameter booster to determine whether to use a tree-based model or a linear model, which divides the space into two subspaces. Moreover, we also combine the two search spaces via an algorithm variable, leading to a CASH problem

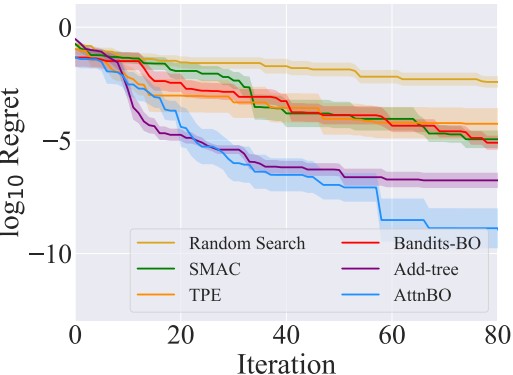

Figure 3: Performance of our AttnBO and baselines on the conditional simulation objective function.

and making the search space more complex having six subspaces and 15 hyperparameters. The details of these search spaces are shown in Appendix C.2.

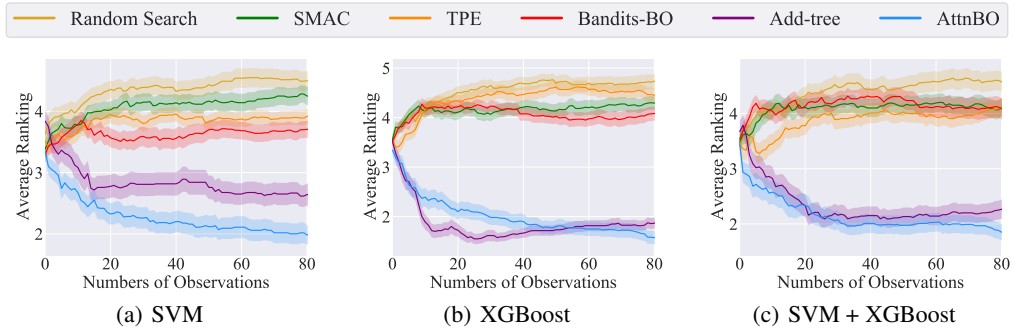

Figure 4: Performance of various black-box optimization methods on three machine-learning benchmarks evaluated on real-world OpenML datasets.

Supported by OpenML (Vanschoren et al., 2013), we consider 6 most evaluated datasets whose task_ids are: [10101, 37, 9967, 9946, 10093, 3494]. Both SVM and XGBoost models can be evaluated on all these tasks.

**Baselines.** We compare AttnBO with Random Search (Bergstra & Bengio, 2012) and four BO baselines for the conditional space on all benchmarks, including two GP-based methods (Bandits-BO (Nguyen et al., 2020), AddTree (Ma & Blaschko, 2020b;a)) and two non-GP methods (SMAC (Hutter et al., 2011), TPE (Bergstra et al., 2011)). Moreover, we also compare with Bandits-BO under a parallel setting on the real-world OpenML benchmarks to demonstrate our ability of batch optimization. The implementation details of these baselines can be found in Appendix B.2.

**Experimental Set-up.** We train the embedding layer and attention-based encoder by maximizing the negative log marginal likelihood according to eq.5 for 100 epochs using Adam optimizer. We set the initial learning rate to 0.01 and reduce it by half every 30 epochs. More details of our implementation can be found in Appendix B.1. For each experiment, following the settings of Bandits-BO, we give $2n$ random points to initialize BO methods. Then, we run BO on the simulation and OpenML tasks until 80 observations (without initial points) are collected and repeat the experiment 10 times in order to reduce the impact of random seeds. For the NAS tasks, we train each candidate on CIFAR-10 training set for 100 epochs and evaluate on the testing set. Because the evaluation of a configuration in this task is very expensive, we only repeat the experiment 3 times.

## 5.1 SIMULATION BENCHMARK

Following the setting of Ma & Blaschko (2020b), we compare our AttnBO with other baselines on this additive structure objective function. As shown in Fig. 3, our method performs best on this simulation benchmark. Here, for a fair comparison, we re-implement the experiment and set the same random seeds as all other algorithms. (Probably, we did not get the same results as shown in their paper due to modifying the number of initial points.) In this task, the objective function is additive as a prior. Although no such prior is added to the model, the attention mechanism can also automatically learn the relationship from the observations.

## 5.2 REAL-WORLD BENCHMARKS ON OPENML

Fig. 4 reports the average ranking of performance on three hierarchical search spaces of two machine-learning models, which were evaluated on 6 real-world datasets randomly selected from OpenML (Vanschoren et al., 2013; Feurer et al., 2019). In our setup, we conducted experiments using three different search spaces: SVM, XGBoost, and the combination of the two shown in Fig. 1. When the search space becomes complex, the gap between different algorithms becomes more obvious, and the performance of the algorithm optimized for the tree structure is significantly improved. We guess this is due to considering the same parameters in different subspaces in the search spaces of SVM and XGBoost. For example, the effect of gamma in RBF is similar to that of sigmoid in SVM. Our proposed method achieves the best performance on all three benchmarks and, in particular, outperforms the start-of-the-art BO method AddTree (Ma & Blaschko, 2020b) for conditional search spaces. We also report the performance of all baselines on each dataset, which can be found in Fig. 15.

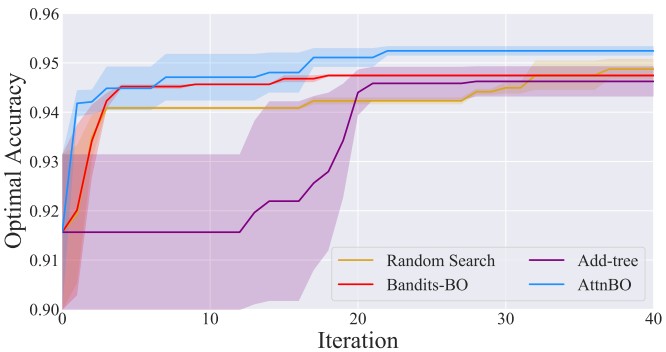

Figure 5: Performance of baselines and AttnBO on the complex NAS space.

### 5.3 Neural Architecture Search

Considering the evaluation of a deep neural network is very expensive, the parallel of BO becomes especially important and necessary, which could improve the efficiency of the optimization process. However, the state-of-the-art method AddTree (Ma & Blaschko, 2020b) is not able to conduct a parallel BO, which will still give only one query per BO iteration in this experiment. We show the optimal accuracy after each BO iteration for all methods in Fig. 5. With more hyperparameters and more complex condition settings, the ability to explore becomes crucial. Compared to other methods, Add-Tree is limited in its capability to explore or exploit various configurations within a BO loop due to the inability to provide batch queries. As a result, the opportunity for observation is reduced, leading to a failure in finding optimal configurations during the early stages. On the other hand, Random Search demonstrates better performance on this task because of its strong ability to explore across each dimension in a larger space with parallelism (Bergstra & Bengio, 2012). In contrast to existing methods, our AttnBO has the advantage of exploiting the relationships between hyperparameters, which allows us to learn better representations of configurations (see Fig. 12). Additionally, AttnBO also enables parallel optimization by selecting the best candidate in each subspace, leading to better performance throughout the BO process.

## 6 Conclusion

In this paper, we proposed a novel attention-based BO framework, named AttnBO, to capture the hierarchical response surface with conditional dependencies by a single GP, which facilitates the application of Bayesian optimization in practical automated machine learning systems. Specifically, we proposed a general embedding method that can introduce the semantic and dependency information into the configurations from different subspaces. Then we utilize an attention-based to capture the relationships among hyperparameters in a configuration and project the configurations from different subspaces, which have different structures and dimensions, into a unified latent space. With the powerful attention-based encoder, we build a single GP model in the latent space and train the parameters of the deep kernel by the negative log marginal likelihood. Moreover, our proposed method can give a batch of quires in a BO iteration, which improves the efficiency when dealing with expensive objective functions. Finally, we conduct the experiments on multiple benchmarks and give sufficient experimental results to demonstrate the effectiveness of our method.

## 7 Broader Impact and Limitations

The proposed method in this paper enables efficient and effective Bayesian optimization in the search spaces that exist dependencies among hyperparameters, which can facilitate the application of Bayesian optimization in practical AutoML systems. We trust our proposed method can be a powerful tool for more complicated AutoML scenarios which include data preprocessing and feature engineering, however, we do not discuss this topic in this paper, which we would pay more attention to this application in our feature work.

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

# A  ALGORITHM

We describe the procedure of our proposed method in Algorithm 1.

---

**Algorithm 1:** AttnBO: An Attention-based Approach for Bayesian Optimization with Dependencies.

---

**Inputs:** A black-box function $f$ defined on a hierarchical search space $\chi = \chi^1 \cup \chi^2 \cup ... \cup \chi^n$
with conditional dependencies;

The name of hyperparameters $name(p_k^{x^i})$ in a subspace $\chi^i$, where $i = 1, 2, ..., n$,

$x^i \in \chi^i \subset \mathbb{R}^{d^i}, k = 1, 2, ..., d^i$;

The batch size $B(B <= n)$;

The number of total training iterations $T$.

1 Randomly sample two initial points $(N^i = 2)$ to evaluate from each subspace, resulting in
$N = 2n$ initial points in total.

2 Get the initial dataset: $D_0 = \left\{ (x_j^i, y_j^i) | i = 1, 2, ..., n, j = 1, 2, ..., N^i \right\}$

3 **for** $t := from\ 1\ to\ T$ **do**

4      Fit the Deep Kernel Gaussian Process by maximizing the log marginal likelihood (eq.5) with

5      Adam optimizer.

6      Optimize the acquisition function in each subspace: $x_*^i = \arg\max_{x \in \chi^i} \alpha(x), i = 1, 2, ..., n$.

7      Get the next queries $X_* = \{x_b | b = 1, 2, ..., B\} = TopB(\{\alpha(x_*^i) | i = 1, 2, ..., n\})$ and their

8      responses: $D_* = \{(x_b, y_b) | b = 1, 2, ..., B, y_b = f(x_b)\}$.

9      Update the dataset of observations $D_t = D_{t-1} \cup D_*$

10 **end**

11 **Output:** The best point $x_{opt}$ in history.

12 $\dagger$: $TopB$ is a function that returns the top B configurations ranked by the acquisition function.

---

# B  IMPLEMENTATION DETAILS

## B.1  ATTNBO

### B.1.1  STRUCTURE-AWARE EMBEDDINGS

We use sequential coding to encode the name of each hyperparameter in a full hierarchical space $\chi$. For example, assume that we have a search space that has three hyperparameters $x1$, $x2$ and $x3$, and $x2$ is a child of $x1$. We create a map to encode $x1$'s name into 1, $x2$'s name into 2 and $x3$'s name into 3. Then, we find the code of each hyperparameter's father node to introduce the dependencies information. Combine the father's name code and its own name code, we can get such codes for the three hyperparameters: $x1 : [0, 1]$, $x2 : [1, 2]$, $x3 : [0, 3]$, where code 0 is the padding code for representing a hyperparameter without father node.

Considering there are some hyperparameters that are lists and have several dimensions, we introduce the dimension information of the hyperparameter. For example, we have two hyperparameters in a neural network search space, which are the number of layers $nums\_layer$ and the number of units per layer $nums\_unit$ respectively. Specifically, $nums\_layer$ is the father node of $nums\_unit$ and ranges from 4 to 7, which indicates that the hyperparameter $nums\_units$ will be a list whose length ranges from 4 to 7 depending on the value of $nums\_layer$. In such a situation, we need to identify each dimension of the hyperparameter using the index of the list because each dimension in this list has the same name and father node. For example, if $nums\_layer$ gets 4, then we can get code 1, 2, 3, 4 for each dimension of $nums\_units$. In addition, if a hyperparameter is a scalar and only has one dimension, we use code 0 to represent its dimension.

Based on these codes, we utilize an embedding layer to get the $name\_emb$ and the father's $name\_emb$, and another embedding layer to get $dim\_emb$ for each hyperparameter. Specifically, we utilize 'nn.Embedding' provided in PyTorch to get the embeddings, which have 64 dimensions in our setting. When we sample a configuration in the search space, we use a linear layer to transform the value of each hyperparameter into a 64-dim vector and concatenate these three embeddings as the

representation of each hyperparameter. Then, we can get the full embedding of the configuration as eq.3 and eq.4 show.

### B.1.2 ATTENTION-BASED ENCODER

We adopt the Transformer encoder as the deep kernel network to project the configurations in different subspaces into a unified latent space $\mathcal{Z}$. Specifically, we employ 6 attention blocks with 2 parallel attention heads. The dimensionality of input and output is dmodel = 256 ($4 \times 64$), and the inner layer also has a dimensionality of 512. We adopt average pooling to integrate the output of the transformer encoder and utilize a multi-layer perceptron (MLP) with 4 hidden layers, which has [128, 128, 128, 32] units of each hidden layer, to project the features of the configurations into 32-dim vectors. In our ablation study, following Dosovitskiy et al. (2021), we utilize another way to integrate the features of the transformer encoder via an extra token, which we named AttnBO-token-mixer in this paper.

### B.1.3 DEEP KERNEL GAUSSIAN PROCESS

For the Gaussian Process model, we utilize Matérn 5/2 as the kernel function and set the mean prior to zero. We adopt the Adam optimizer to train the parameters of the kernel by maximizing the log-likelihood, embedding layer, and attention-based encoder for 100 epochs. We set the learning rate to 0.001 with a decay rate of 0.5 every 30 epochs. For the acquisition, we utilize EI to balance the exploration and exploitation and utilize the lbfgs optimizer to optimize EI in each subspace during the acquisition stage. Unfortunately, a large number of subspaces will make it impossible to optimize EI in each subspace using lbfgs, which performs best in our experiment. In this situation, we can use Thompson sampling as the acquisition to find the next query like Nguyen et al. (2020). If you still want to use EI, you can just simply use random sampling to optimize EI in the full search space.

### B.2 BASELINES

In this section, we provide the specific details of each baseline mentioned in the paper:

**Random Search (RS).** Following the description in Bergstra & Bengio (2012), we sample candidates uniformly at random.

**Tree Parzen Estimator (TPE).** Bergstra et al. (2011) adopt kernel density estimators to model the probability of configurations with bad and good performance respectively. We use the default settings provided in hyperopt package (`https://github.com/hyperopt/hyperopt`).

**SMAC.** Hutter et al. (2011) adopt random forest to model the response surface of the black-box function. When dealing with the search space with dependencies, SMAC imputes the inactive hyperparameters in each subspace with default values. We use the default settings given by scikit-optimize package (`https://github.com/scikit-optimize/scikit-optimize`) and impute the default values as SMAC3 package (`https://github.com/automl/SMAC3`).

**Bandits-BO.** Nguyen et al. (2020) builds a sub-GP in each subspace and uses a Thompson sampling scheme that helps connect both multi-arm bandits and GP-BO in a unified framework. We implement this method in our own framework. For each sub-GP, we use the same settings as our AttnBO except for the deep neural network. We use the Matérn 5/2 as the kernel function and fit the sub-GPs using slice sampling.

**AddTree.** Ma & Blaschko (2020b) proposed an Add-Tree covariance function to capture the global response surface using a single GP, which is the state-of-the-art BO method for the hierarchical search spaces. We use the default settings provided by `https://github.com/maxc01/addtree`.

## C DETAILS OF THE BENCHMARKS AND EXPERIMENTS

To better display the search space with dependencies, we define a YAML format to represent the search space. Following Xue et al. (2023), we adopt the keywords "type" and "range" to represent

the type and domain of the hyperparameter respectively. In addition, we also define the keyword "submodule" to indicate the dependencies among hyperparameters. As for dependencies, in this search space format, we support two types. When the number or distribution of one parameter depends on another parameter, we can use the keyword "submodule" to indicate the relationship between these parameters. For the type of each hyperparameter, we support choice, int, and float for the categorical, integer, and decimal hyperparameters respectively. As to the range of integer hyperparameters, we adopt the left-closed and right-open intervals to represent. For example, if an integer hyperparameter $x1$ has the range [0...2], it can be 0 or 1. For every search space, we will give both the YAML-style and figure-style representation.

## C.1 SIMULATION BENCHMARK

The tree-structure search space of the simulation function that was originally presented in Jenatton et al. (2017) consists of 9 hyperparameters as Listing 1 and Fig. 6 shows.

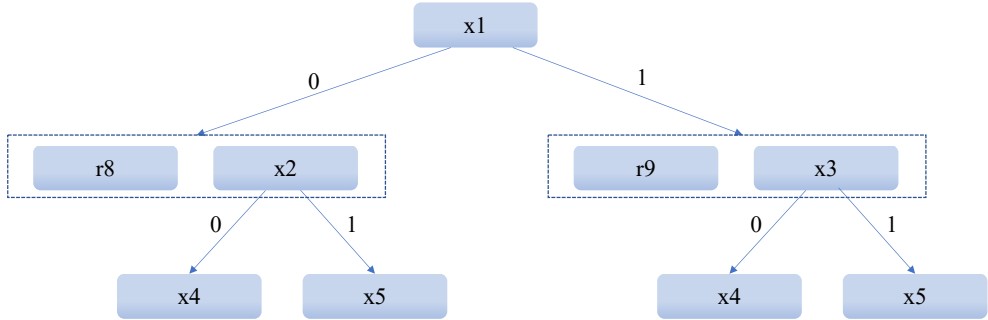

Figure 6: The tree-structured search space on the simulation function presented in Jenatton et al. (2017).

Listing 1: YAML-style representation of the simulation search space.

```
x1:
  type:  choice
  range: {0, 1}
  submodule:
    0:
      r8:
        type: int
        range: [0...2]
      x2:
        type:  choice
        range: {0, 1}
        submodule:
          0:
            x4:
              type:  float
              range: [-1...1]
          1:
            x5:
              type:  float
              range: [-1...1]
    1:
      r9:
        type: int
```

```
      range: [0...2]
   x3:
      type:  choice
      range: {0, 1}
      submodule:
        0:
          x6:
             type:  float
             range: [−1...1]
        1:
          x7:
             type:  float
             range: [−1...1]
```

## C.2 OPENML BENCHMARKS

We define two search spaces with dependencies for two popular machine-learning algorithms (SVM and XGBoost) and evaluate the configurations on 6 most evaluated datasets whose task_ids are: [10101, 37, 9967, 9946, 10093, 3494]. Furthermore, we compose the two search spaces into a more complex CASH space to further explore the capabilities of our method. In this section, we will give the details of the three search spaces and show the details of the experimental results for each search space on all datasets in Fig. 15.

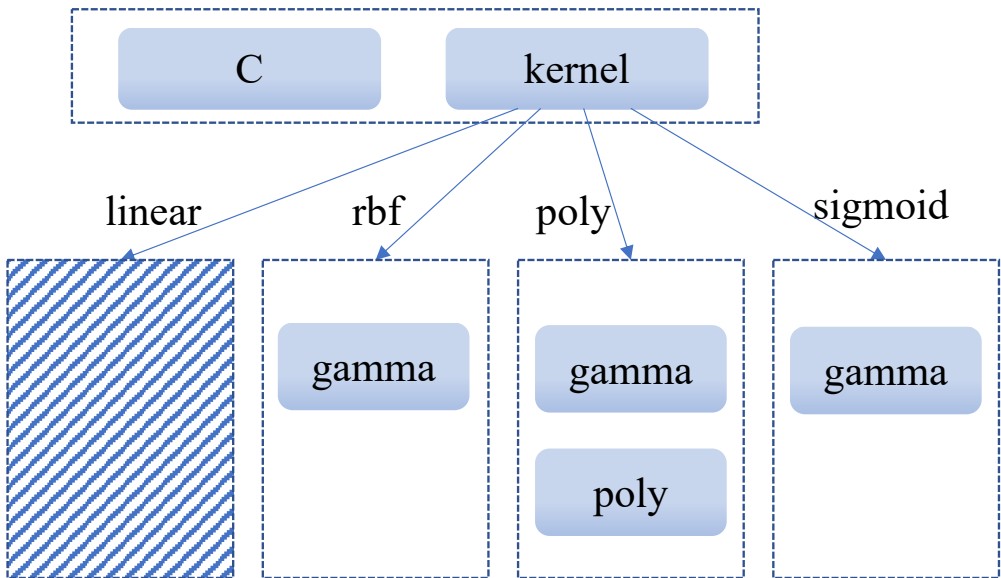

Figure 7: The tree-structured search space of SVM on the tabular classification tasks. When the kernel is linear for the SVM model, the shaded box indicates that there are no hyperparameters in this case.

### C.2.1 SVM SEARCH SPACE

The structure of the SVM search space is shown in Listing 2 and Fig. 7. When the kernel is set to linear, there is no extra hyperparameter and no "submodule" in the YAML file.

Listing 2: YAML-style representation of the SVM search space.

```
C:
  type: float
  range: [0.001...1000]
kernel:
  type: choice
  range: {"linear", "poly", "sigmoid", "rbf"}
  submodule:
    poly:
      degree:
        type: int
        range: [2...6]
      gamma:
        type: float
        range: [0.001...1000]
    sigmoid:
      gamma:
        type: float
        range: [0.001...1000]
    rbf:
      gamma:
        type: float
        range: [0.001...1000]
```

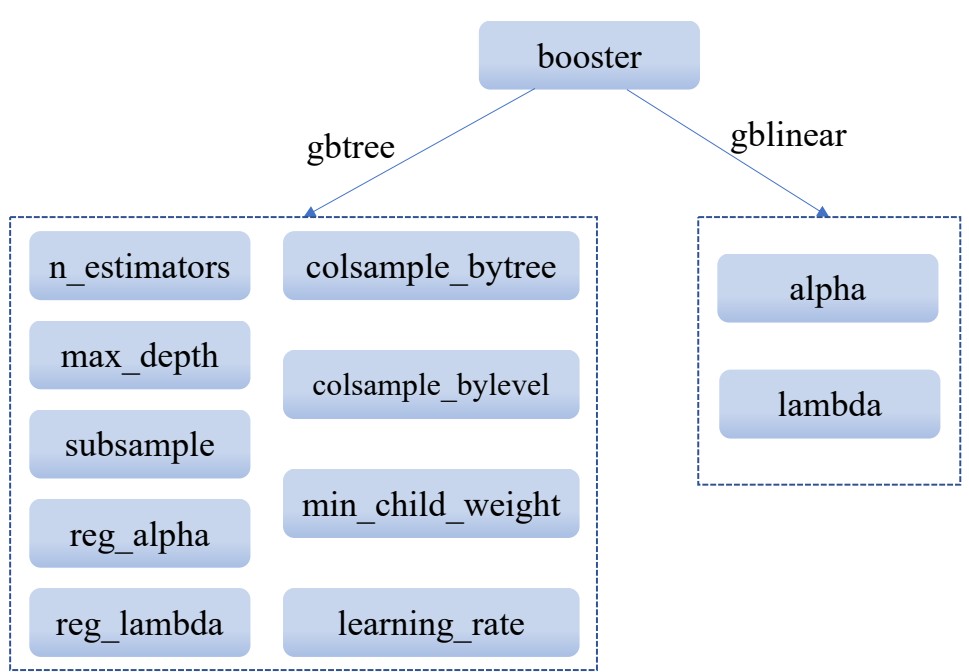

Figure 8: The tree-structured search space of XGBoost on the tabular classification tasks.

### C.2.2 XGBoost Search Space

The XGBoost search space consists of 10 hyperparameters and is more complex than the SVM search space. Its structure can be seen in Listing 3 and Fig. 8.

Listing 3: YAML-style representation of the XGBoost search space.

```
booster:
  type:   choice
  range: {gbtree, gblinear}
  submodule:
    gbtree:
      n_estimators:
        type:   int
        range: [50...501]
      learning_rate:
        type:   float
        range: [0.001...0.1]
      min_child_weight:
        type:   float
        range: [1...128]
      max_depth:
        type:   int
        range: [1...11]
      subsample:
        type:   float
        range: [0.1...0.999]
      colsample_bytree:
        type:   float
        range: [0.046776...0.998424]
      colsample_bylevel:
        type:   float
        range: [0.046776...0.998424]
      reg_alpha:
        type:   float
        range: [0.001...1000]
      reg_lambda:
        type:   float
        range: [0.001...1000]
    gblinear:
      reg_alpha:
        type:   float
        range: [0.001...1000]
      reg_lambda:
        type:   float
        range: [0.001...1000]
```

### C.2.3   SVM + XGBOOST SEARCH SPACE

In order to further explore the capabilities of our method, we compose the two search spaces into a more complex CASH space by introducing a meta-level hyperparameter "algorithm" to choose which algorithm will be used to evaluate. The structure of the composed CASH search space is shown in Listing 4 and Fig. 9.

Listing 4: YAML-style representation of the SVM + XGBoost search space.

```
algorithm:
  type:   choice
  range: {xgboost, svm}
  submodule:
    xgboost:
      booster:
        type:   choice
```

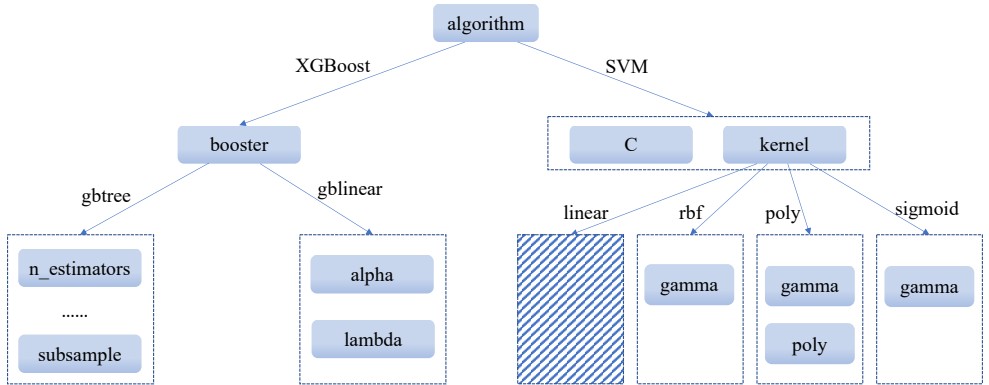

Figure 9: The tree-structured search space on the tabular classification tasks. When the kernel is linear for the SVM model, the shaded box indicates that there are no hyperparameters in this case.

```
      range: {gbtree, gblinear}
    submodule:
      gbtree:
        n_estimators:
          type:  int
          range: [50...501]
        learning_rate:
          type:  float
          range: [0.001...0.1]
        min_child_weight:
          type:  float
          range: [1...128]
        max_depth:
          type:  int
          range: [1...11]
        subsample:
          type:  float
          range: [0.1...0.999]
        colsample_bytree:
          type:  float
          range: [0.046776...0.998424]
        colsample_bylevel:
          type:  float
          range: [0.046776...0.998424]
        reg_alpha:
          type:  float
          range: [0.001...1000]
        reg_lambda:
          type:  float
          range: [0.001...1000]
      gblinear:
        reg_alpha:
          type:  float
          range: [0.001...1000]
        reg_lambda:
          type:  float
          range: [0.001...1000]
  svm:
    C:
```

```
type:    float
range:   [0.001...1000]
kernel:
  type:    choice
  range:  {"linear", "poly", "sigmoid", "rbf"}
  submodule:
    poly:
      degree:
        type:    int
        range:   [2...6]
      gamma:
        type:    float
        range:   [0.001...1000]
    sigmoid:
      gamma:
        type:    float
        range:   [0.001...1000]
    rbf:
      gamma:
        type:    float
        range:   [0.001...1000]
```

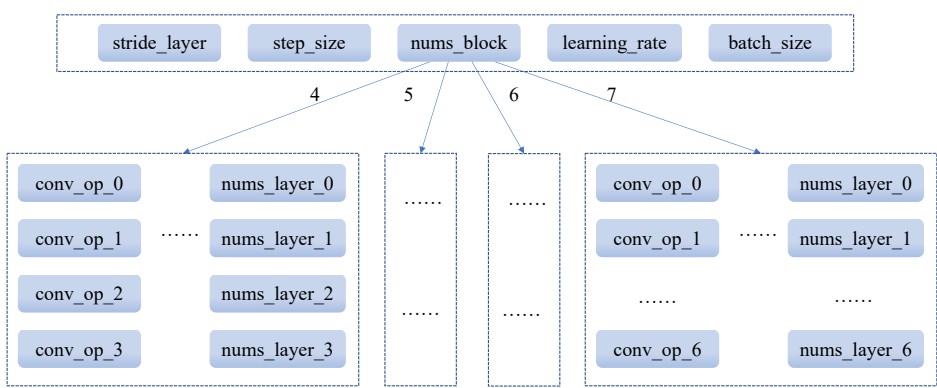

Figure 10: The tree-structured search space of the NAS task evaluated on CIFAR-10 dataset.

## C.3    Neural Architecture Search

Following Tan et al. (2019), we define a factorized hierarchical search space to find the best network architecture and its training configurations. The search space consists of two aspects: 1) Neural network architectures. 2) Hyperparameters of the optimizer used for training the neural networks.

As shown in Fig. **?**, for the network architectures, we group the network layers into a number of provisioned skeletons, called blocks, based on some solid works (Howard et al., 2017; Sandler et al., 2018; Tan et al., 2019; Tan & Le, 2019) in computer vision. Each block contains various repeated identical layers, except striding. Only the first layer has stride 2 if the block needs to downsample, while all other layers have stride 1. We use the hyperparameter "stride_layer" to control this operation. For each block, we search for the types of stacked convolution operations and connections for a single layer and the number of layers "nums_layer"($N$), and then for one layer $i$ is repeated $N_i$ times (e.g., Layer 4-1 to 4-N_4 are the same, where N_4 represents the number of repeated layers in the 4th block). Now we describe the details of each hyperparameter:

1. $nums\_block$. The number of blocks.

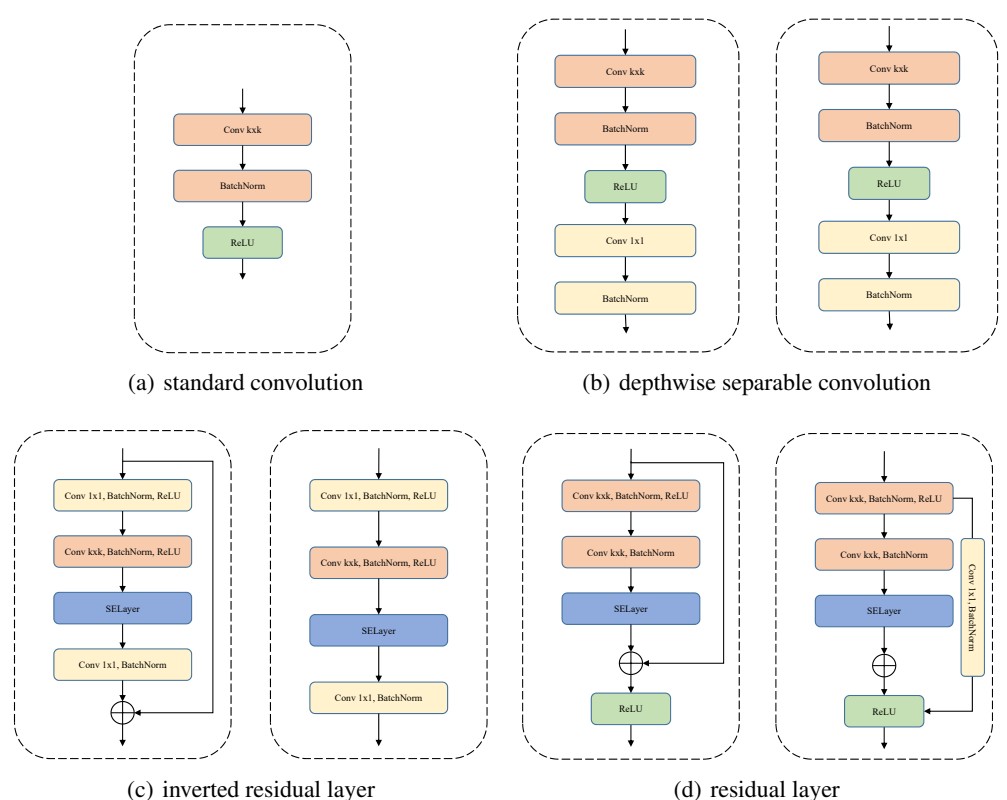

Figure 11: Searched architectures for NAS task.

2. *conv_op*. The convolution operation type for a single layer of each block. In our settings, following Tan et al. (2019), there are 4 provisioned types available, represented by codes 0, 1, 2, and 3 respectively. 1) The first is the standard convolution layer (Simonyan & Zisserman, 2015), which consists of a 2D convolution operation with a kernel size of ($kernel\_size \times kernel\_size$), a batch normalization operation and a ReLU activation function. 2) The second type is the depthwise separable convolution layer (Howard et al., 2017). It has the same function as the standard convolution layer but is more efficient, which is a form of factorized convolutions with a standard convolution into a depthwise convolution and a 1×1 convolution called a pointwise convolution. 3) The next one is the inverted residual layer (Sandler et al., 2018), where each layer contains an input followed by two bottlenecks and two expansion layers between them. 4) The last type is the ResNet layer commonly used in computer vision tasks (He et al., 2016).

3. *kernel_size*. The size of the convolution kernel in one convolution block.

4. *nums_layer*. The number of layers in each block.

5. *expend_ration*. The ratio for expending, if using the inverted residue block (Sandler et al., 2018).

6. *seratio*. The ratio of squeezing and expending if containing such structure.

7. *nums_channel*. The number of channels for each block.

8. *stride_layer*. The number of strides for each block is represented in binary.

The optimization hyperparameters The details of each hyperparameter are as follows:

1. *learning_rate*. The learning rate determines the speed of the network's training and convergence.

2. *step_size*. Size of the change in the parameter when the optimizer updates the parameter.

3. *batch_size*. Batch size determines how many data points will be used for training in each iteration.

The structure of the search space of the NAS task is shown in Listing 5 and Fig. 10.

# D   VISUALIZATION OF ATTENTION MAPS

To give some insights into the relationships between the hyperparameters, we visualize the attention maps learned on the NAS task where $nums\_block = 4$ under two different random seeds in Fig. 12. In order to preserve the original practical meaning of each embedding, we only visualize the average of different parallel heads in the first attention block. Note that the attention scores in each row are normalized by the softmax function. As for the neural architecture, we observed that the first three blocks are more important for achieving good performance. Additionally, we found that hyperparameters related to optimization, such as $batch\_size$ and $learning\_rate$, are highly correlated with various hyperparameters of the network architecture, which is intuitive in this task.

Listing 5: YAML-style representation of the NAS search space.

```yaml
# hyperparameters of the network architecture
nums_block:
  type: int
  range:
  - 4...8
  submodule:
    conv_op:
      type: choice
      range: {0, 1, 2, 3}
    expand_ratio:
      type:  int
      range:  [5...7]
    seratio:
      type: choice
      range: {0, 8, 16}
    kernel_size:
      type: choice
      range: {3, 5}
    nums_layer:
      type: choice
      range: {0, 1, 2}
    nums_channel:
      type: choice
      range: {1, 1.25, 1.3}
stride_layer:
  type: choice
  range: {43, 44}

# hyperparameters for optimization
learning_rate:
  type: float
  range: [0.07...0.15]
step_size:
  type: int
  range: [70...90]
batch_size:
  type: powerint2
  range: [5...8]
```

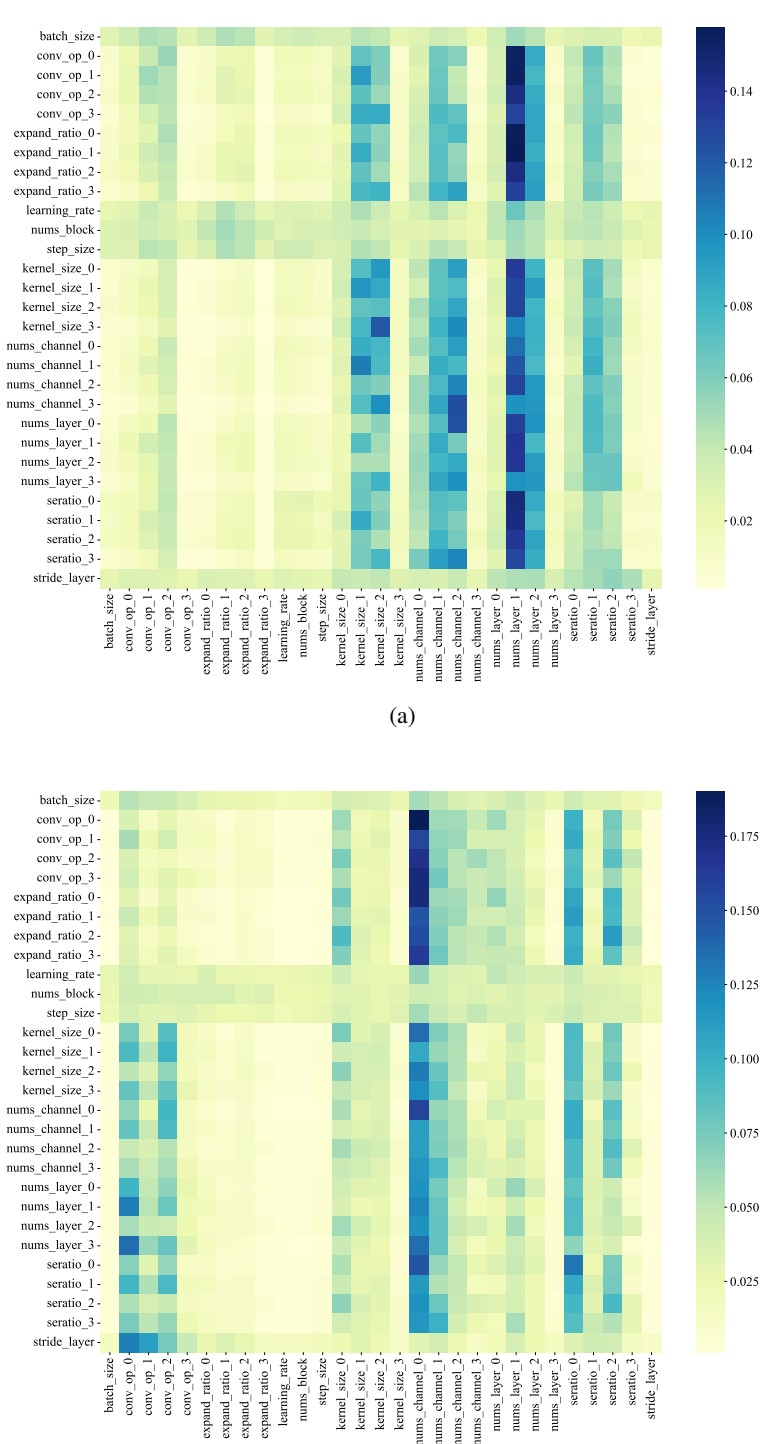

(a)

(b)

Figure 12: Visualization of attention maps for all tasks.

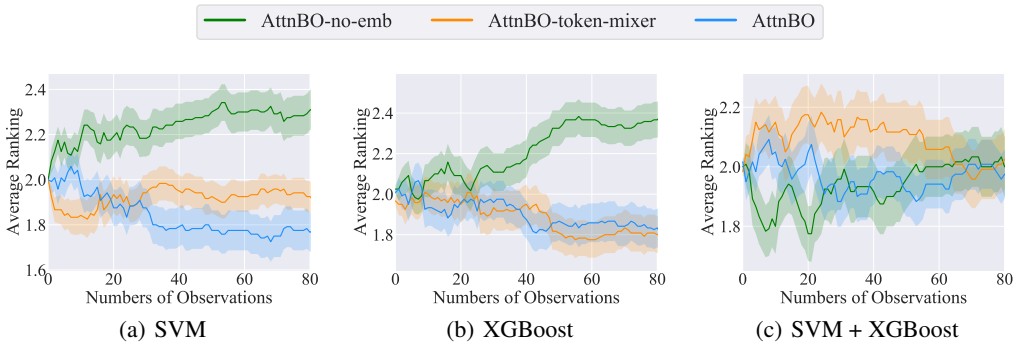

Figure 13: Performance of our AttnBO and two variants.

# E    ABLATION STUDY

To verify the effectiveness of the embedding method we proposed, we conduct ablation studies on the machine-learning benchmarks. Specifically, we compare the attention-based encoder with-/without our proposed structure-aware embeddings and the results are shown in Fig. 13. In this experiment, we compare two different embedding methods, AttenBO-no-emb, and our method. We try a naive approach of embedding configurations through values and directly concatenating them into a sequence, dubbed AttenBO-no-emb. The proposed embedding method facilitates the attention module to learn dependencies in a tree-structured search space in separate search spaces of SVM and XGBoost, compared with naive methods. In combinatorial search spaces, as the complexity of the large search space increases, the performance difference becomes smaller and the variation becomes larger. However, the graph shows a downward trend, and we guess that the algorithm needs more iterations to converge in complex search spaces, the experiment results are shown in the supplementary. Furthermore, we validate two methods commonly used in Transformer architectures, classification token style (AttnBO-token-mixer) and average output embeddings style (Hwang et al., 2022) (AttnBO). From experiments, we find that the latter performs better than the former. Obviously, our proposed embedding method helps to capture the relationships between hyperparameters and leads to higher effectiveness.

In addition, the results for the complex search spaces are shown in Fig. 14. It can be seen that our algorithm converges in more iterations and outperforms the baseline methods, which indicates the attention module captures the relationships between hyperparameters and leads to higher effectiveness.

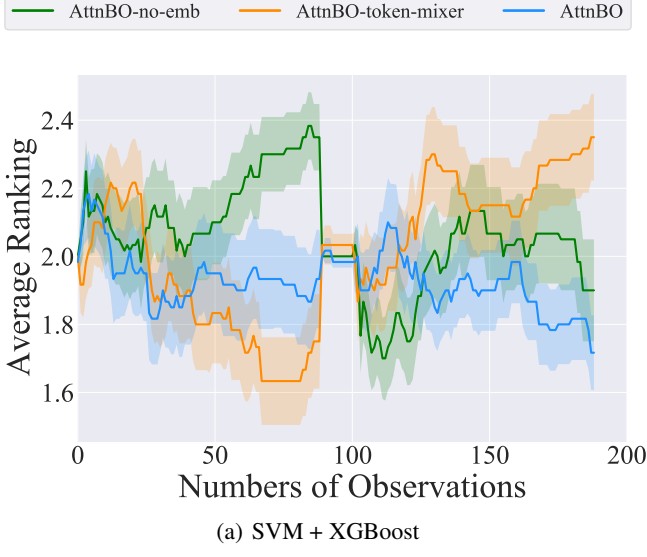

(a) SVM + XGBoost

Figure 14: Performance of our AttnBO and two variant methods.

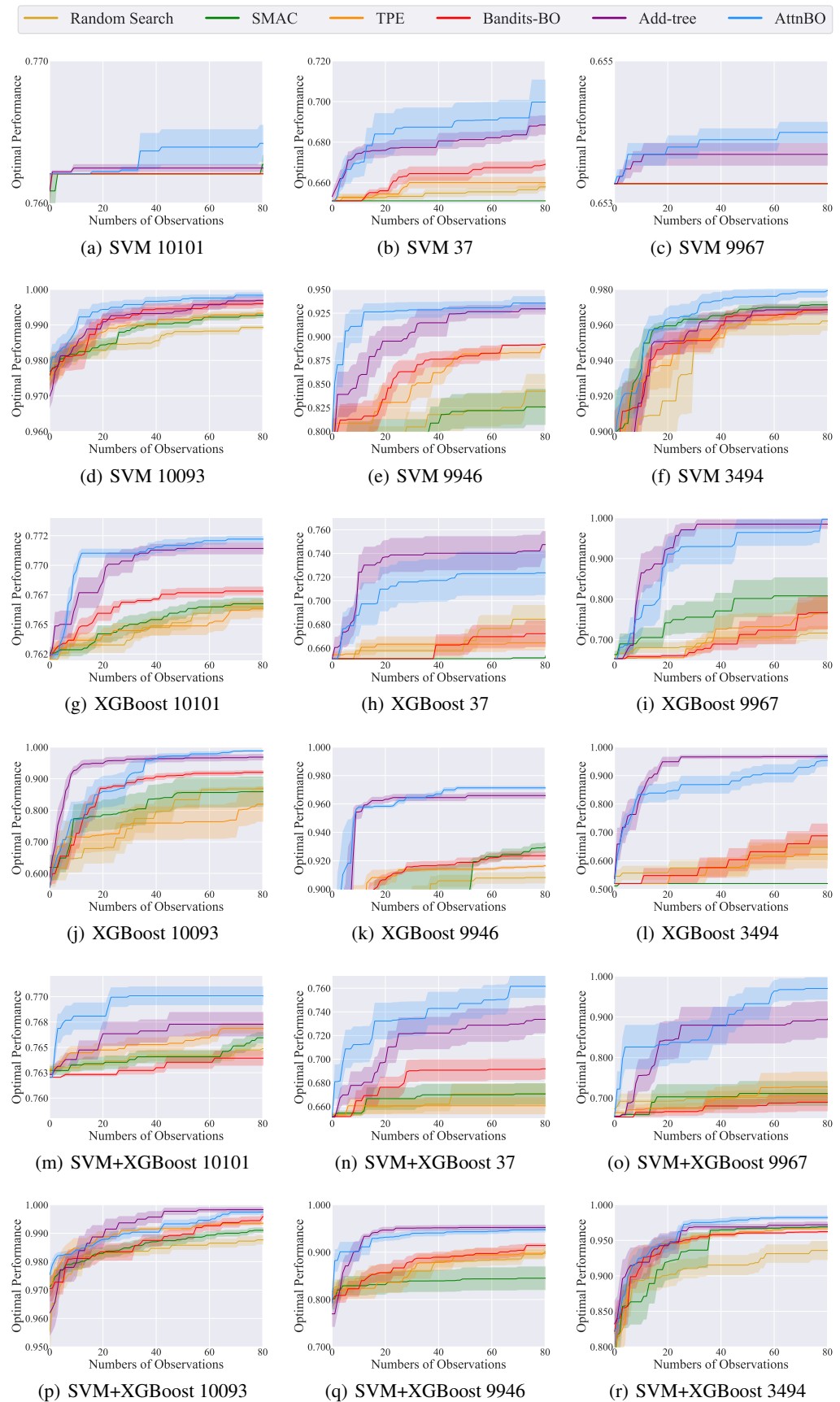

Figure 15: Performance of various black-box optimization methods on three machine-learning benchmarks evaluated on real-world OpenML datasets.

