# OpenReview forum: "An Attention-based Approach for Bayesian Optimization with Dependencies"
_ICLR.cc/2024/Conference — ICLR 2024 Conference Withdrawn Submission_

### Official Review · Reviewer_FWug · 2023-10-30

**Soundness:** 2 fair
**Presentation:** 1 poor
**Contribution:** 2 fair
**Rating:** 1
**Confidence:** 4

**Summary:**

This paper tackles a Bayesian optimization problem with conditional search spaces.  Using the multi-head attention component, which is used for the Transformer network, the proposed method carries out deep kernel learning, and then the learned kernel is employed to select the next query point following a Bayesian optimization procedure.  Finally, the authors provide experimental results to verify the effectiveness of the proposed method.

**Strengths:**

* It solves an interesting topic in Bayesian optimization.
* Application of attention mechanism seems effective for the deep kernel learning in Bayesian optimization.
* Experimental results demonstrate the effectiveness of the proposed method.

**Weaknesses:**

* Writing should be improved.  There are many typos and grammar errors.  For example,

1. n noisy observations in Page 1 should be $n$ noisy observations.
1. $d$ is the dimension in Page 1 should be $d$ is the number of dimensions.
1. even different dimensions in Page 1 should be the different number of dimensions.
1. configurations in the same subspace has in Page 1 should be configurations in the same subspace have.
1. black-box object function in Page 2 should be black-box objective function
1. (SE) kernel function in Page 2 should be (SE) kernel functions.
1. quires in Page 2 should be queries.
1. cifer-10 in Page 7 should be CIFAR-10.
1. quires in Page 9 should be queries.

I think there might be other cases.  These should be carefully fixed.

* Important references, for example, [1, 2, 3], are missing.

[1] Lyu, Wenlong, et al. "Efficient Bayesian Optimization with Deep Kernel Learning and Transformer Pre-trained on Multiple Heterogeneous Datasets." arXiv preprint arXiv:2308.04660 (2023).

[2] Bowden, J., et al. Deep kernel Bayesian optimization. No. LLNL-CONF-819001. Lawrence Livermore National Lab.(LLNL), Livermore, CA (United States), 2021.

[3] Chowdhury, Sankalan Pal, et al. "Learning the Transformer Kernel." Transactions on Machine Learning Research (2022).

Many papers including the references mentioned have studied Transformer-based or attention-based kernel learning.  These are all related to this work.  Unfortunately, the absence of the discussion on this line of research degrades the quality of this work.  Moreover, the novelty of this work should be also debatable.

* A method to select a query point by considering a tree-structured search space is questionable.  Eventually, query points are determined by optimizing the respective acquisition function defined on subspaces according to Algorithm 1.  I think it reduces the need to use attention-based kernels.  I think there might exist a way to optimize an acquisition function on the entire conditional search space.

* The scale of experiments are relatively small.

**Questions:**

* The sentence "provides a theoretical regret bounds" in Page 1 needs references.  Moreover, the theoretical analysis for the expected improvement is limited, unlike GP-UCB.
* I am not sure that the sentence "However, these existing methods do not readily extend to search spaces containing both categorical and numerical hyperparameters in a complex, structured relationship" in Page 2 is true.  There is previous work that considers the setting of both categorical and numerical parameters.
* For Equation (2), $\phi$ is parameters?  It seems like a function, not parameters, according to Equation (2).
* Could you elaborate the sentence "However, not only do different algorithms require different hyperparameters, but also different microstructures have different value ranges for the same hyperparameter" in Page 4?
* What is a booster?  I think it should be a base learner or base estimator.
* What is a father vertex or father node?  Is it different from a parent node?
* I think that "to the best of our knowledge, there is no suitable hand-crafted kernel function capable of modeling the similarity between these configurations from different subspaces which have different dimensions and dependencies" is not true according to the references I mentioned early.
* I would like to ask about how the name and value of hyperparameters are handled.  Is a name processed as a natural language?  Is a value is normalized?
* For Figures 3 and 4, the results at iteration 0 are different across methods.  Did you fix the random seeds for initialization?

---

### Official Review · Reviewer_8zXp · 2023-10-31

**Soundness:** 2 fair
**Presentation:** 2 fair
**Contribution:** 2 fair
**Rating:** 5
**Confidence:** 4

**Summary:**

The paper consider the problem of optimizing black-box functions over conditional/heirarchical search spaces. The key idea is to use a deep kernel Gaussian process as the surrogate model on top of a attention module for embedding the inputs into a common latent space. The inputs are augmented with few identifiers before passing to the attention layer. Experiments are performed on a simulation benchmark, NAS benchmark, and hyper-parameter tuning benchmarks from OpenML.

**Strengths:**

- The paper considers an important problem with many real-world applications.

- Investigating attention based neural network modules for handling conditional search spaces is quite interesting and deserves appreciation.

**Weaknesses:**

- The main contribution of the paper is to include tree structure aware identifiers with each hyperparameter (Appendix B.1.1) and learning embeddings with the attention module. However, these identifiers are a hand-designed feature and only an artifact of the way tree search space is constructed. Since one of the major goals of the paper is AutoML, this seems to be defeating the main purpose by adding manual feature construction. Please clarify and discuss this important point.

- Some statements in the paper are not entirely incorrect. For example, it is mentioned that "there is no suitable hand-crafted kernel function capable of modeling the similarity between these configurations from" but arc-kernel (reference [1] below) is one such kernel that handles conditional search spaces and is a natural baseline for this problem. Please consider reevaluating them.

    - [1] Swersky, K., Duvenaud, D., Snoek, J., Hutter, F., & Osborne, M. A. (2014). Raiders of the lost architecture: Kernels for Bayesian optimization in conditional parameter spaces. arXiv preprint arXiv:1409.4011.

- I am afraid that Add-tree baseline (which is significantly easier to implement) outperforms the proposed approach in multiple instances in Figure 4 and Figure 15. Please explain this. I also think a simple baseline with imputing missing values for inactive hyperparameters is a simple baseline that is easy to try should be included. If possible, please consider including it.

**Questions:**

Please see weaknesses section above. I am more than happy to increase my score if the questions are answered appropriately.

---

### Official Review · Reviewer_4yMm · 2023-10-31

**Soundness:** 2 fair
**Presentation:** 2 fair
**Contribution:** 2 fair
**Rating:** 3
**Confidence:** 4

**Summary:**

The paper introduces a novel approach for addressing the hierarchical search space in Bayesian optimization. This approach involves a tree-structure-oriented embedding to represent the search space and employs a deep kernel based on an attention-oriented encoder to capture the structure within the Gaussian process in Bayesian optimization. Empirical results presented in the paper provide evidence of the effectiveness and efficiency of the proposed embedding and the attention mechanism within the kernel.

**Strengths:**

1. The paper is generally well-structured, conveying essential concepts through figures and visual representations of structures.

2. The paper introduces a novel embedding based on tree structures to incorporate the known hierarchical structure of the objective in Bayesian optimization. It demonstrates the effectiveness of this approach with both general empirical studies against baselines on various tasks and ablation studies.

3. Additionally, the paper integrates recent advancements in the attention mechanism from natural language processing into Bayesian optimization, connecting these two distinct domains.

**Weaknesses:**

1. The most concerning problem is the motivation and problem setup. While the author argues that the proposed embedding and modeling are designed to deal with specific hierarchical dependencies, there is neither a formularized statement of the problem involving the hierarchical structure (only a short footnote in the introduction and an example which is illustrated in Figure 1) nor a clear definition or statement of considered 'hyperparameters' (it seems that the paper deal with the anisotropy of the objective yet this is not clearly defined). The essential concepts need to be more specific throughout the paper. Otherwise, the vagueness undermines the soundness and contribution of the paper.

2. The discussion on related work is limited. For example, the recent advancement in casual-model-based BO methods [1,2] that also explicitly deals with the dependencies on the search space needs to be included.

3. Some results are placed in the appendix but are actually critical while insufficient.  The ablation study doesn't show consistent improvement over AttnBO-no-emb in both Figure 13-c and Figure 14-a. Apart from the ablation study, the visualization of the attention maps only uses two random seeds and, therefore, lacks significance.

4. The baselines in the experiment are insufficient compared to the related methods discussed in Section 2. Specifically, Naïve DKL-based BO or VAE-based BO are not compared as baselines. This comparison could serve as the ablation study of the proposed encoder-based AttenBO.

5. The paper's primary contribution resides in the domain of representation and kernel learning, offering a novel solution for the Gaussian process in the context of structured input. The acquisition function is not tailored to this particular setting. The proposed method should be discussed within the broader scope of GP studies rather than being confined to a Bayesian optimization-oriented approach. In that sense, the intermediate evidence of improved GP regression is far from sufficient.

**Reference**

[1] Aglietti, Virginia, Xiaoyu Lu, Andrei Paleyes, and Javier González. "Causal Bayesian optimization." In International Conference on Artificial Intelligence and Statistics, pp. 3155-3164. PMLR, 2020.

[2] Sussex, Scott, Anastasiia Makarova, and Andreas Krause. "Model-based causal Bayesian optimization." arXiv preprint arXiv:2211.10257 (2022).

**Questions:**

1. Could the author comment on how deep kernel learning converges with limited training data, especially given that overfitting is a known problem in DKL when trained with log-likelihood as training loss [3]?

2. Could the author discuss the impact on the optimization performance if the proposed embedding is applied with the classical DKL instead of the attention-oriented structure and offer corresponding empirical evidence?


**Reference**

[3] Ober, Sebastian W., Carl E. Rasmussen, and Mark van der Wilk. "The promises and pitfalls of deep kernel learning." In Uncertainty in Artificial Intelligence, pp. 1206-1216. PMLR, 2021.